# Effects of a Mixed Exercise Program on Overweight and Obese Children and Adolescents: A Pilot, Uncontrolled Study

**DOI:** 10.3390/ijerph19159258

**Published:** 2022-07-28

**Authors:** Roberto Pippi, Gabriele Mascherini, Pascal Izzicupo, Vittorio Bini, Carmine Giuseppe Fanelli

**Affiliations:** 1Healthy Lifestyle Institute, C.U.R.I.A.Mo. (Centro Universitario Ricerca Interdipartimentale Attività Motoria), Department of Medicine and Surgery, University of Perugia, Via G. Bambagioni, 19, 06126 Perugia, Italy; carmine.fanelli@unipg.it; 2Department of Experimental and Clinical Medicine, University of Florence, 50134 Florence, Italy; gabriele.mascherini@unifi.it; 3Department of Medicine and Aging Sciences, University “G. D’Annunzio” of Chieti-Pescara, 66100 Chieti, Italy; pascal.izzicupo@unich.it; 4Department of Medicine and Surgery, University of Perugia, Via Gambuli, 1, 06132 Perugia, Italy; vittorio.bini@unipg.it

**Keywords:** body composition, strength, obesity, physical activity

## Abstract

Pediatric excess weight has reached severity worldwide, affecting physical health. Decreasing weight and body mass index (BMI) after exercise intervention reduces the cardiometabolic consequences; the role of age and gender on the effectiveness of exercise in overweight youth was debated in this study. A total of 138 overweight/obese young (75 girls, 63 boys) were recruited at Perugia (Italy) University to follow an exercise program. Participants were allocated into two groups (children, *n* = 88 and adolescents, *n* = 50). The study aimed to verify the efficacy of a mixed resistance–endurance exercise program in anthropometric and physical performance measures, evaluating the influence of gender and age on two groups of young overweight/obese participants. In children, we observed a statistically significant improvement in fat mass percentage, fat-free mass, waist circumference (WC), fat mass, as well as in strength, endurance, speed, and flexibility measures. We also observed reduced WC and waist-to-height ratio (WHtR) values in girls. In the adolescents’ subgroup, results showed a statistically significant variation in fat mass percentage, BMI, WC, and WHtR, and strength of the upper and lower limbs; we also observed a weight reduction in girls. A clinical approach, with the combination of strength and dynamometric tests plus the body composition study using air plethysmography methodology, is health-effective and allows for the monitoring of the efficacy of an exercise program in overweight/obese young people.

## 1. Introduction

Pediatric excess weight has reached worldwide prevalence and severity, with adverse effects on physical health [1]. For example, a high-fat mass percentage increases the risk of hyperlipidaemia, hypertension, and insulin resistance and promotes a long-term chronic inflammatory state [2]. As a result, excess adipose tissue increases the risk of chronic cardiometabolic diseases, such as type 2 diabetes, metabolic syndrome, coronary heart disease, and stroke [3,4].

Multidisciplinary interventions are considered effective in reducing cardiometabolic risk factors in overweight children and adolescents. These approaches mainly focus on weight reduction and lifestyle modification [5]. It has been shown that cardiometabolic outcomes are significantly reduced with the decrease in weight and body mass index (BMI) following interventions based on modifying eating habits and physical activity levels. Specifically, exercise interventions in overweight children improve body composition by lowering body fat, blood sugar, and waist circumference (WC) [4]. 

Historically, exercise programs have focused primarily on aerobic cardiovascular fitness workouts for overweight youth. However, more recent evidence suggests including resistance exercise programs combined with endurance or using resistance programs alone [6]. For example, progressive resistance training (RT) improvements in muscle strength, muscle power, and local muscle endurance in children and adolescents have been well described [7]. Thus, the World Health Organization recently recommended activities strengthening muscle and bone for young people [8]. Additionally, RT has produced many health-related benefits, including improvements in cardiovascular fitness, body composition, bone mineral density, blood lipid profiles, and insulin sensitivity [9]. Moreover, hand grip strength (HG) is used to predict health throughout an individual’s lifespan [10,11] (i.e., cardiovascular death, bone fragility, and sarcopenia) and, in children and adolescents, is significantly associated with current and future health status [12]. 

Systematic reviews indicate that concurrent aerobic plus resistance training is more effective than interventions based on a single modality on a range of physical, physiological, and psychological parameters in children and adolescents [13,14]. However, several factors, such as ponderal status, sex, maturation, and basal fitness may affect the intervention effectiveness. For example, a recent study by Enríquez-del-Castillo et al. (2022) shows how the state of maturation can play a role in modulating changes in physical fitness following an exercise program in children [15]. However, the difference between different age groups during youth and the influence that gender has on the effectiveness of an exercise program in overweight young people is currently unclear. For instance, it has been shown that participating in a summer training camp can elicit different responses in the activity levels of obese male and female children [16]. This information would allow for highly individualized training programs to be carried out, obtaining maximum responsiveness from the proposed training stimuli.

Therefore, this study aims to verify the efficacy of a mixed resistance-endurance exercise program and evaluate the influence of gender and age on effectiveness in two groups (children and adolescents) of overweight young people, using the combination of strength and dynamometric tests plus an aerial plethysmography methodology for the study of body composition.

## 2. Materials and Methods

### 2.1. Participants

Between 2010 and 2017, C.U.R.I.A.Mo. has carried out clinical activity mainly with adult patients (around 1000/year). In recent years, the clinical care model has been validated in young subjects. For young people, a much smaller number was reserved (around 50/year) than for the adults.

From October 2013 to May 2017, a total sample of 138 (Figure 1) overweight/obese young people (75 girls aged 11.55 ± 2.61 years and 63 boys aged 11.14 ± 2.72) were recruited at the C.U.R.I.A.Mo. (Centro Universitario Ricerca Interdipartimentale Attività Motoria) The Healthy Lifestyle Institute of the University of Perugia (Perugia, Italy) follows a multidisciplinary, family-based intervention [17]. Participants were allocated into two groups according to baseline age: the children’s group, aged 6–12 yrs (*n* = 88, aged 9.70 ± 1.60) and the adolescents’ group, aged 13–17 yrs (*n* = 50, aged 14.28 ± 1.28). All baseline values are reported in Table 1. Inclusion criteria were (1) the presence of all anthropometric and physical performance data collected both before and after the intervention; (2) age between 6 and 17 years, (3) BMI over 85° percentile [18], and (4) the adherence percentage ≥50% at the exercise sessions. The exclusion criteria were (1) the absence of the parents’ informed written consent to the intervention, (2) the presence of musculoskeletal disorders or any potential medical contraindications to exercise, and (3) family difficulties in managing the timetable of the intervention activities.

### 2.2. Intervention

This study followed the C.U.R.I.A.Mo. Healthy Lifestyle Institute clinical, curative intervention for subjects with overweight/obesity, previously described in detail by some authors [17,19,20,21]. Study participants underwent one individual medical examination conducted by a pediatrician and an intensive gym-based exercise intervention program, as described in more detail in Section 2.2 (Table 2). 

Additionally, study subjects participated in nutritional and psychological counselling sessions that involved all the family members’ components. Briefly, individual nutritional counselling (not prescriptive) sessions and educational group sessions were provided to explain strategies to improve the adherence to the principles of the Mediterranean diet, according to the Italian Standard Treatments of Childhood Obesity (SIO) [22,23]. The psychological counselling sessions were characterized with a family-based approach and aimed to involve both parents in counselling focused on the family’s needs. Previous studies have already reported the results of these components [17,19].

Using a quasi-experimental study design, the effectiveness of the exercise program was established by comparing the baseline parameters (T0) with those at the end (T1) of the exercise program. 

The study was conducted in compliance with the guidelines in the Declaration of Helsinki and the C.U.R.I.A.Mo. project has been registered in the Australian New Zealand Clinical Trials Registry (a Primary Registry in the WHO registry network) with the number: ACTRN12611000255987. All the participants’ parents gave their informed consent in written form to participate in the study. 

#### 2.2.1. Exercise Intervention 

Based on studies where exercise programs are described [24,25], it was decided to administer different exercise programs (Figure 2) in terms of length, according to the age of the participants. In fact, at the beginning of our clinical experience, we observed that three months of activity in the gym were enough for adolescents to obtain significant learning and results (unpublished data). In contrast, for the younger ones, it was necessary to repeat the quarter of activity at least once. Workouts were ideated according to current scientific guidelines [26,27].

*Exercise intervention for children.* Children were involved in a six-month exercise intervention program, with two sessions per week lasting 90 min each. Physical activities for this group were organized in small groups of a minimum of six and a maximum of 12 participants. Two certified exercise specialists supervised all the training sessions, consisting of 15 min of warm-up, 50 min of calisthenics, mixed exercises to improve conditional and coordination skills, a group game phase (15 min), and stretching (10 min). During the training sessions, exercises were proposed to improve general fitness and physical abilities, including spatial coordination, aerobic capacity, flexibility, and muscle strength. Each central phase of the exercise sessions aimed to work more on specific motor objectives. However, we worked in a multilateral way in each session, using play as the primary medium. The exercises consisted of combinations of basic motor gestures such as crawling, walking, running, jumping (in a different direction), climbing, rolling, throwing, and grabbing, using small tools such as hoops, cones, small balls, and balls. Every week modified versions of the games/exercises were proposed, with a gradual increase in effort. Final stretches were performed through the active stretching method, involving muscular groups used in the session (i.e., triceps surae, hamstrings, quadriceps femoris, etc.). Participants performed each stretching exercise helpless and statically and were instructed to hold each position for 20 s.

*Exercise intervention for adolescents*. The adolescents’ group participated in a three-month exercise intervention, with two sessions per week, each lasting 90 min. Participants had no previous resistance training experience. For this reason, two weeks were dedicated to familiarizing the participants with the exercise techniques and the isotonic machines. Activities were organized for 5–6 participants/group, with one certified exercise specialist supervising. Each session included 60 min of cardiovascular workouts (using ergometers such as, i.e., Run 500, Recline 600 XT Pro, and Top 600 XT Pro and Synchro, Technogym, Cesena, Italy), 30 min of circuit training for muscular strength (using free lifts and isotonic machines, such as, i.e., Lat Machine, Chest Press, Leg Press, Technogym, Cesena, Italy). According to participants’ fitness and adaptation, exercise intensity was gradually increased by c.a. 5% every six sessions, starting from 50% of heart rate reserve (HRR) for the aerobic phase and 55% of 1 RM for the muscular phase strength work. Each participant wore a Polar FT4 © Heart Rate Monitor (Polar, Finland) and communicated their instant heart rate every 30 s to the specialist who supervised the training and recorded the data in a special digital log.

### 2.3. Measures 

The percentage of adherence to exercise intervention was calculated as the number of sessions performed/total number of sessions × 100.

#### 2.3.1. Anthropometric Variables Measures

Height, weight, body composition (as Fat mass in % and Fat-free mass in kg), BMI, WC, and waist-to-height ratio (WHTR) were assessed during a medical examination conducted by a pediatrician with a supporting dietician. Height was measured twice to the nearest centimetre in the upright position by a Portable Stadiometer [28]. Participants’ body weight and composition were evaluated through Air Displacement Plethysmography (ADP) using a gold standard instrument BOD POD (BOD POD^®^ Composition System; COSMED, Albano Laziale, Italy). The subject’s mass was measured using a digital scale integrated with BOD POD instrument. The instrument and its scale were calibrated before every measurement [29]. Before the measurements, the participants were asked to have been fasting for at least 4 h; to have abstained from exercise for at least 12 h; to have an empty bladder; and to have abstained from alcohol for at least 48 h. Taking into account the schedule of the Centre and considering the participants’ school commitments, these measurements were carried out in the afternoon, trying as much as possible to schedule the tests and retests in the same time slot. BMI was estimated according to the equation (weight in kilograms (kg)/(height (m)^2^). WC was measured with the participant standing after expiration, using Seca Ergonomic Girth measuring tape with an automatic roll-up (Seca 201, Seca North America, Chino, CA, USA). Finally, WHTR was calculated as WC divided by height [30]. 

#### 2.3.2. Physical Performance Measures

In all samples, muscular strength was measured through Handgrip (HG) strength test for the right and left hand, conducted using a handgrip dynamometer (Dynex, Akern, Firenze, Italy) that recorded the strength changes of each participant. They were asked to tighten the dynamometer as much as possible until they were told to stop by the exercise specialist. The test was repeated on both right and left hands to assess different grip strengths. For each test session (T0 and T1), we made three measurements of the maximum force value with each hand; finally, the average value was taken. Furthermore, flexibility was estimated through the Sit & Reach test performed from a standing position (vertical flexion, VB) and from a sitting position (horizontal flexion, HB). We asked participants to flex the torso with the arms outstretched and to reach gradually the closest position to the tiptoes. We recorded the distance reached by the participants’ fingertips, using a 40 cm × 40 cm sit and reach box (Technogym, Cesena, Italy).

In the children’s group, the strength of the lower and upper limbs was also assessed by two submaximal functional tests, the Sargent test (ST) and the medicine ball throw (MBT), executed forward and backward. For the first test, the participants were asked to stand in front of a wall and stretch their arms as high as possible, and the exercise specialist recorded the measurements achieved. Afterward, the children were asked to perform a vertical jump. Finally, the difference in centimeters between the two signs of the height reached from a standstill with outstretched arms and that obtained after the jump was considered. Then, in the MBT test, the children were asked to sit down holding a medicine ball with two hands stretched over their heads and, after extending/flexing the body, to throw the ball as far back/forward as possible, based on the two different tests.

To assess aerobic capacity (VO_2_), the 6 Minute Walk Test (6MWT) was used [31]. For this test, we recorded the distance a child can walk, as quickly as possible, in 6 min. Finally, to assess speed (recorded in seconds and hundredths), a 30 m speed test was performed. The children had to run the distance of 30 m in line as fast as possible, and the time measurement was carried out with a stopwatch.

In the adolescents’ group, the isotonic machine test was used to assess the strength of the lower (leg press and leg extension submaximal test values) and upper (lat machine and chest press submaximal test values) limbs. The Brzycki equation was used to determine the value of 1 RM [32]. Aerobic capacity was measured through the Balke protocol test (please see the Fitmate user manual) conducted on a treadmill, setting to a speed of 5.3 km/h for men and 4.3 km/h for women, with an incline of 0 degrees (0%), using indirect calorimetry with a gas exchange analyzer (Fit Mate, Cosmed, Rome, Italy) [33]. 

A more detailed description of the procedure used in the tests mentioned above is given by Ranucci et al. [17]. 

### 2.4. Statistical Analysis

We presented descriptive analyses of means ± standard deviations, and/or percentages for each variable at baseline (T0). 

A Univariate Analysis of Variance test was run to compare all variables across the gender category at baseline. Then, to evaluate the effects of the intervention, a repeated-measures multivariate analysis of variance was used, using gender group category as a between factor. Moreover, a paired *t*-test for delta changes (T1 vs. T0) was executed for both children and adolescents’ subgroups and boys and girls. *p* values ≤ 0.05 were set as statistically significant, and the effect size was measured using partial eta-squares. All analyses were conducted through SPSS^®^ Software, version 25.0 (IBM Corp. Released 2017. IBM SPSS Statistics for Windows, Version 25.0. Armonk, NY, USA: IBM Corp.).

### 2.5. Sample Size Calculation

A sample size of 138 achieves 88% power to detect a difference between the null hypothesis mean of 0.0 and the alternative hypothesis mean of −3.5 cm in WC difference with an estimated standard deviation of 13.0 cm and with a significance level (alpha) of 0.05 using a two-sided one-sample *t*-test.

## 3. Results

### 3.1. Baseline Ressults 

*Children’s group.* Anthropometric variables measures values showed no differences between gender subgroup values at baseline. 

For the physical performance measures values, we observe differences in VB and HB: girls at T0 were more flexible than boys (both in vertical, *p* < 0.001, and horizontal position, *p* = 0.001).

*Adolescents’ group*. In anthropometric variables, we observed some differences in weight (*p* = 0.001), fat-free mass (*p* < 0.001), BMI (*p* = 0.038), and WC (*p* = 0.037), with higher values in males than females.

In physical performance measures values, boys’ values are higher than girls’ values for lat machines (*p* = 0.001), chest press (*p* < 0.001), and leg extension (*p* = 0.038). Finally, boys’ VO_2_ max mean values are statistically significant (*p* = 0.007) higher than girls’. All the baseline results are reported in Table 1.

### 3.2. Exercise Program Effects

After the exercise program, the percentage of adherence to exercise intervention was 61.5 ± 27.63 for the entire sample. In the children group, we observed a percentage of 54.98 ± 26.42 (boys 55.82 ± 25.84, girls 54.17 ± 27.38; *p* = 0.813), while in the adolescents we had 69.69 ± 27.20 (girls 61.70 ± 30.6, boys 80.48 ± 17.24; *p* = 0.018). 

*Children’s group.* In anthropometric variables measures values, we observed an improvement in all the anthropometric variables measures values, except for body weight (*p* = 0.092). For the physical performance measures values, we measured a statistically significant increase in all the variables, except the HG test for the left hand. All the results are reported in Table 2. Moreover, subgrouping this sample for sex, in the children’s subgroup (Table 3a), we observed a statistically significant improvement both for boys and girls in height, fat mass percentage and fat-free mass (*p* < 0.001), WC, and fat mass (*p* <0.05), MBT forward, backward, ST and 6MWT (*p* < 0.05). Additionally, we found a significant variation in WHtR (*p* = 0.029) value in girls and VB (*p* = 0.008) in boys. 

*Adolescents’ group*. In anthropometric variables, we observed statistically significant improvements (*p* < 0.005) in all the measures values, while for the physical performance measures values, we observed an increase in strength values but not in bending (horizontal, *p* = 0.165; vertical, *p* = 0.145) and V0_2_ max values (*p* = 0.169). In the adolescents’ subgroup (Table 3b), results showed a statistically significant variation both for boys and girls in fat mass percentage, BMI, WC, and WHtR (*p* < 0.05), Lat, Chest, Leg, and Legext (*p* < 0.001). Moreover, significant improvements were observed for girls’ weight (*p* = 0.021) and fat mass (*p* < 0.001), and for height (*p* = 0.003), HG for right (*p* = 0.013) and left (*p* = 0.033) hands.

## 4. Discussion

This study aimed to present the effects of a mixed exercise program (endurance plus resistance training) on two groups of overweight/obese young people. Moreover, we observed the influence of sex and age on the program’s effectiveness. 

All sample data demonstrate the effectiveness of the exercise program, resulting in an improvement in all the anthropometric (except for body weight) and physical performance measures values. 

Previous research supports that the multidisciplinary approach is one of the most valid instruments to reduce overweight and obesity in children [34,35] and adolescents, especially when exercise is included [17,22,36].

Bharath et al. [37], exploring the efficacy of mixed exercise on visceral adiposity in a group of adolescent girls with obesity, showed that aerobic plus resistance exercise reduced metabolic risk factors in obese adolescents. In line with these results, our study observed an improvement in fat mass, WC, and WHTR in both the boys’ and girls’ groups. In addition, other authors [4] that studied the effects of mixed exercise in children showed that exercise interventions significantly improved several cardiometabolic risk factors, such as BMI. However, it did not significantly influence WC and body weight, as reported by other authors [17,38]. Therefore, our results are not entirely in accord with them. Our study showed an improvement in WC but not in BMI and weight in the children’s group, probably due to the variations in height that occur at this age. Furthermore, we hypothesize that weight is not a specific variable that describes the changes in body composition caused by exercise. 

Previous studies [39,40] underlined the importance of body composition evaluation in pediatric clinical practice. However, anthropometric measures such as BMI and bodyweight alone have insufficient sensitivity for overweight and obesity treatment and management. Moreover, fat-free mass evaluation allows for the tailoring of an exercise treatment. Our study did not observe statistically significant variations in body weight in either the children’s (nor the boys’ or girls’) or adolescents’ boys. Moreover, the BMI mean values of the children’s group did not show significant variations. Given these results, according to Adom et al., which claimed the importance of the duration of interventions on anthropometric variables [41], we hypothesize that the duration of our intervention is perhaps too short. Future studies of longer duration are needed to clarify these aspects, or it may be useful to influence the frequency and intensity of the workouts. Nevertheless, according to a previous study [42], we observed a significant decrease in cardiovascular risks resulting from the improved sample mean values of WHTR index and WC, variables associated with cardiovascular risk. Particularly, in the children group at T1, we recorded WHTR index mean values near 0.5, which is the clinical cut-off [43,44,45].

Evidence suggests that objective physical performance and ability measures may predict subsequent health problems [46]. This aspect is essential, particularly in childhood, the stage of life where we lay the foundations for health status in adulthood and old age. Our work adopted the C.U.R.I.A.Mo. consolidated assessment methodology, well verified in children and adolescents [17]. We used an objective assessment of muscular strength with the HG strength test for the right and left hand, using a dynamometer. This measurement is a quick and inexpensive way to measure muscular strength that [10] in adolescents is an emerging risk factor for major causes of death in young adulthood.

A previous study showed conflicting opinions regarding the effects of an exercise program on the strength levels of overweight/obese female adolescents. Although some authors reported that children and adolescents with obesity showed impaired muscular fitness compared to normal-weight peers [47,48], other authors reported positive exercise effects on muscular strength [49]. In line with the latter results, we observed an improvement in the dynamic muscular strength performance of children and adolescents with overweight/obesity. 

Although some authors reported that males are usually stronger than females in all age groups [50,51], there are no definitive indications. Other authors have not shown significant differences between boys and girls [52]. Moreover, a previous study showed significant differences in HG test mean values between children and adolescents of different ages [50,53]. Our study observed an increase in mean values in boys and girls in the children’s group. In our opinion, although not statistically significant, these results may be clinically relevant. According to Thivel et al. [47], improving muscular fitness in children with obesity is critical because it supports their physical autonomy and adherence to exercise-based weight management intervention. 

Previous studies focused on physical performance measures in young people as they are the expression of motor abilities [54] and individual biological changes (i.e., growth and fitness level) [55,56]. We reported a significant increase in height in the children’s and adolescent boys’ groups. In addition, our results, which were in accord with those of some authors that have studied school populations [57], showed significant improvements both in muscular strength and speed in boys and girls. Notably, as found in Magnani et al. [58], we observed improvements in strength values, as measured by the Sargent Test, both in boys and girls. 

### Strength and Limitations 

This study presents many objective variables and measurements of performance and body composition, according to the well-established C.U.R.I.A.Mo. evaluation model. The same operator at T0 and T1 always conducted measurements and tests to minimize operator variability. Further, to avoid bias, the operators who conducted the training program did not perform the tests on the young people they had followed but were joined by operators who had followed the training of other groups. As specified in Section 2.2.1, the results were obtained with two different exercise interventions for children and adolescents, as for methodological and technical reasons, the same type of exercise intervention could not be proposed to both groups.

Additionally, all operators followed a rigorous protocol based on the guidelines. Finally, this study presents the results that derive from a robust methodology. Air plethysmography (gold standard) was used for body composition assessment, while a dynamometer measured muscle strength. Unfortunately, these measurements are not very common in children and adolescents. In particular, air plethysmography is not an inexpensive analysis. 

This study is a pilot study involving small groups of children and adolescents; therefore, no definitive conclusions can be based on this data. While it is true that it is difficult to obtain a satisfactory degree of adherence to treatment in clinical care programs, this is even more so when it is necessary to involve entire families. Moreover, this uncontrolled study was based on the C.U.R.I.A.Mo. clinical model. In this context, all participants were referred by their pediatrician to the C.U.R.I.A.Mo. Institute to receive care and were allocated to the treatment groups; for these reasons, this study did not provide a control group. For these reasons, further studies with larger study populations and control groups will be needed to confirm and generalize the results of this pilot study. Finally, this study did not include nutritional and psychological measures, which are very important in subjects with overweight/obesity during childhood.

## 5. Conclusions

This mixed exercise program was demonstrated health-effective for two groups of children and adolescents with overweight/ obesity. In the children group, data showed an improvement in all the anthropometric (except for body weight) and physical performance measures values (except for the HG test for left hand) in children and boys. In girls, we observed reduced WC and WHtR values linked to cardiovascular risk. In the adolescents’ group, this exercise intervention improved anthropometric variables values (even those associated with a reduction in cardiovascular risk) and increased strength values in males and females. In the girls’ subgroup, the exercise program also resulted in a decrease in weight. A multidimensional approach based on the combination of strength tests and the study of body composition through ADP can represent an effective tool for controlling the health and assessing the efficacy of an exercise program in young people with overweight/obesity. Further studies with larger study populations will be needed to confirm and generalize the results of this pilot study.

## Figures and Tables

**Figure 1 ijerph-19-09258-f001:**
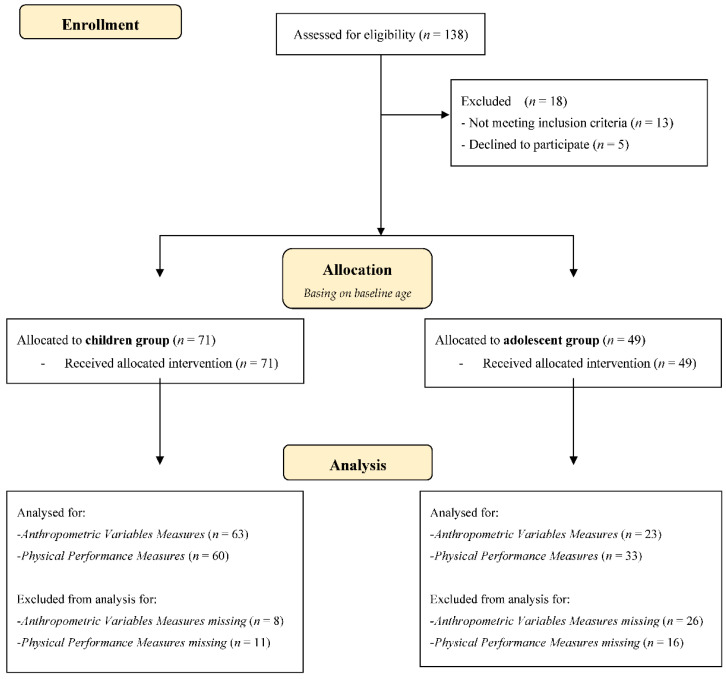
CONSORT 2010 Flow Diagram.

**Figure 2 ijerph-19-09258-f002:**
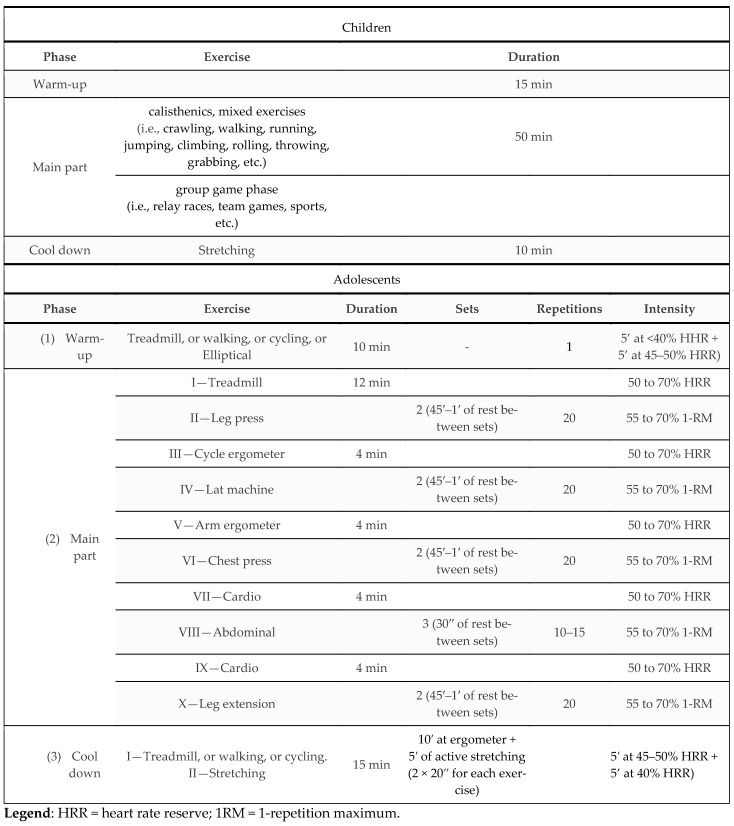
Examples of exercises for both children and adolescents’ intervention.

**Table 1 ijerph-19-09258-t001:** Anthropometric and Physical Performance measures baseline values in children and adolescent groups. Data are presented as mean (Mn) ± Standard Deviation (SD) values. Statistical significance was set for *p* values ≤ 0.05.

	**Children**	**Gender Group Category**			
		**Girls**	**Boys**	**Boys vs. Girls**	
	**Mn ± SD**	**Mn ± SD**	**Mn ± SD**	**F**	** *p* **
Weight (kg)	61.64 ± 13.33	64.32 ± 11.54	58.73 ± 14.65	3.219	0.077
Fat mass (%)	40.06 ± 4.99	40.48 ± 3.79	39.61 ± 6.07	0.534	0.468
Fat-free mass (kg)	25.10 ± 7.34	26.12 ± 5.76	23.99 ± 8.70	2.154	0.147
BMI (kg/m^2^)	28.81 ± 3.8	28.77 ± 2.41	28.85 ± 4.05	0.11	0.916
WC (cm)	94.09 ± 8.58	94.12 ± 8.50	94.05 ± 8.91	0.001	0.980
WHTR	0.63 ± 0.04	0.63 ± 0.05	0.64 ± 0.04	0.197	0.660
MBT forward (cm)	4.49 ± 1.02	4.53 ± 1.04	4.45 ± 1.02	0.103	0.749
MBT backward (cm)	4.70 ± 1.34	4.79 ± 1.46	4.62 ± 1.23	0.223	0.639
ST (kg)	20.28 ± 6.80	22.04 ± 7.25	18.64 ± 6.02	3.524	0.066
HG test for right hand (kg)	16.48 ± 4.51	16.90 ± 5.21	15.79 ± 3,14	0.556	0.461
HG test for left hand (kg)	14.53 ± 3.42	14.15 ± 3.32	15.24 ± 3.63	0.841	0.365
6 MWT (m)	712.27 ± 96.41	717.25 ± 99.40	707.13 ± 94.52	0.177	0.676
30 mt speed (s)	1.13 ± 4.33	6.69 ± 1.13	7.00 ± 1.12	1228	0.272
VB (cm)	−3.42 ± 7.97	1.09 ± 6.21	−8.06 ± 6.89	31.707	<0.001
HB (cm)	32.36 ± 7.83	35.81 ± 6.31	29.04 ± 7.80	12.506	0.001
	**Adolescents**	**Gender Group Category**			
		**Girls**	**Boys**	**Boys vs. Girls**	
	**Mn ± SD**	**Mn ± SD**	**Mn ± SD**	**F**	** *p* **
Weight (kg)	90.12 ± 17.52	83.23 ± 14.89	99.30 ± 16.79	12.520	0.001
Fat mass (%)	39.05 ± 7.55	40.20 ± 6.42	37.51 ± 8.78	1.541	0.221
Fat-free mass (kg)	53.43 ± 9.36	48.53 ± 6.06	59.73 ± 9.17	25.874	<0.001
BMI (kg/m^2^)	32.88 ± 5.09	31.58 ± 4.99	34.61 ± 4.80	4.562	0.038
WC (cm)	106.79 ± 12.42	103.52 ± 11.76	111.26 ± 12.19	4.615	0.037
WHTR	0.65 ± 0.07	0.64 ± 0.07	0.66 ± 0.07	1.283	0.264
Lat machine (kg)	36.20 ± 9.54	32.05 ± 5.14	40.97 ± 11.23	11.707	0.001
Chest press (kg)	28.54 ± 9.43	24.06 ± 6.04	33.70 ± 10.10	14.877	<0.001
Leg press (kg)	168.26 ± 53.34	156.78 ± 39.52	181.46 ± 64.32	2.365	0.132
Leg extension (kg)	34.79 ± 13.33	30.94 ± 12.42	39.45 ± 13.21	4.608	0.038
HG test for right hand (kg)	22.90 ± 8.93	20.22 ± 10.34	26.34 ± 5.70	1.969	0.182
HG test for left hand (kg)	20.64 ± 8.59	17.99 ± 9.06	24.05 ± 7.14	2.106	0.169
VB (cm)	−6.06 ± 9.53	−3.89 ± 12.09	−8.22 ± 6.02	0.927	0.350
HB (cm)	29.28 ± 9.75	31.11 ± 10.78	27.44 ± 8.85	0.622	0.442
VO_2_ max (mL/min/kg)	28.96 ± 5.25	26.05 ± 3.89	32.19 ± 4.77	9.540	0.007

**Legend:** BMI = body mass index; WC = waist circumference; WHtR = waist-to-height-ratio; LAT = lat machine test value; CHEST = chest press test value; PRESS = leg press test value; LEGEXT = leg extension test value; HG = Hand Grip test value; VB = vertical bending test value; HB = horizontal bending test value; VO_2_ max = maximal oxygen consumption value.

**Table 2 ijerph-19-09258-t002:** Repeated Measure Multivariate Analysis of Variance to analyze differences in all variables in children and adolescent groups.

**Children**	**T0**	**T1**	**Time T0 vs. T1**		**Time * Gender Group Category**	
	**Mn ± SD**	**Mn ± SD**	** *p* **	**Partial η^2^**	** *p* **	**Partial η^2^**
Weight (kg)	62.04 ± 12.15	62.94 ± 11.76	0.092	0.046	0.310	0.017
Fat mass (%)	40.29 ± 4.46	36.63 ± 5.38	<0.001	0.507	0.559	0.006
Fat-free mass (kg)	36.79 ± 7.08	39.62 ± 6.86	<0.001	0.482	0.217	0.025
BMI (kg/m^2^)	28.59 ± 2.80	27.97 ± 2.84	0.012	0.106	0.790	0.001
WC (cm)	94.62 ± 8.15	91.03 ± 7.88	<0.001	0.449	0.935	<0.001
WHtR	0.64 ± 0.04	0.53 ± 0.21	0.004	0.261	0.882	0.001
MBT forward (cm)	4.52. ± 1.04	4.98 ± 1.07	<0.001	0.326	0.584	0.005
MBT backward (cm)	4.76 ± 1.36	5.48 ± 1.71	<0.001	0.388	0.554	0.007
ST (kg)	20.83 ± 7.21	24.22 ± 7.64	<0.001	0.340	0.637	0.005
HG test for right hand (kg)	16.65 ± 4.57	18.15 ± 5.43	0.042	0.113	0.175	0.052
HG test for left hand (kg)	14.40 ± 3.54	15.33 ± 5.13	0.214	0.048	0.560	0.011
6 MWT (m)	715.34 ± 97.86	775.50 ± 126.91	<0.001	0.371	0.392	0.013
30 mt speed (s)	6.80 ± 1.18	6.19 ± 0.95	<0.001	0.577	0.650	0.004
VB (cm)	−3.48 ± 8.22	−1.81 ± 8.16	0.006	0.127	0.123	0.041
HB (cm)	32.41 ± 8.05	34.16 ± 8.39	0.016	0.117	0.778	0.002
**Adolescents**	**T0**	**T1**	**Time T0 vs. T1**		**Time * Gender Group Category**	
	**Mn ± SD**	**Mn ± SD**	** *p* **	**Partial η^2^**	** *p* **	**Partial η^2^**
Weight (kg)	90.23 ± 16.52	88.13 ± 17.51	0.014	0.255	0.646	0.10
Fat mass (%)	40.57 ± 6.83	36.97 ± 7.41	<0.001	0.526	0.198	0.078
Fat-free mass (kg)	52.96 ± 7.13	54.83 ± 8.58	0.020	0.233	0.119	0.112
BMI (kg/m^2^)	32.71 ± 5.41	31.67 ± 5.78	<0.001	0.461	0.791	0.003
WC (cm)	105.913 ± 12.80	101.87 ± 11.25	0.001	0.445	0.956	<0.001
WHtR	0.64 ± 0.08	0.61 ± 0.07	<0.001	0.497	0.764	0.004
Lat machine (kg)	38.23 ± 9.69	46.82 ± 11.09	<0.001	0.780	0.320	0.032
Chest press (kg)	30.77 ± 9.50	40.33 ± 12.51	<0.001	0.733	0.214	0.049
Leg press (kg)	174.73 ± 57.26	215.48 ± 66.13	<0.001	0.548	0.732	0.004
Leg extension (kg)	37.10 ± 12.53	54.13 ± 14.70	<0.001	0.712	0.691	0.005
HG test for right hand (kg)	23.18 ± 8.60	18.05 ± 6.15	0.016	0.370	0.407	0.054
HG test for left hand (kg)	21.52 ± 8.13	16.54 ± 5.94	0.018	0.359	0.367	0.063
VB (cm)	−6.06 ± 9.53	−4.22 ± 10.51	0.145	0.128	0.964	<0.001
HB (cm)	29.28 ± 9.75	30.61 ± 10.40	0.165	0.117	0.553	0.022
VO_2_ max	28.97 ± 5.40	30.16 ± 6.11	0.169	0.115	0.852	0.002

**Legend:** BMI = body mass index; WC = waist circumference; WHtR = waist-to-height-ratio; LAT = lat machine test value; CHEST = chest press test value; PRESS = leg press test value; LEGEXT = leg extension test value; HG = Hand Grip test value; VB = vertical bending test value; HB = horizontal bending test value; VO_2_ max = maximal oxygen consumption value.

**Table 3 ijerph-19-09258-t003:** (**a**) Post-intervention assessments: Anthropometric and Physical Performance measures values in the children subgroup. Results are presented as Δ (T1 − T0), means, and SDs. Statistical significance was set for *p* values ≤ 0.05. (**b**) Post-intervention assessments: Anthropometric and Physical Performance measures values in the adolescents’ subgroup. Results are presented as Δ (T1 − T0), means, and SDs. Statistical significance was set for *p* values ≤ 0.05.

**(a)**						
	**Boys**			**Girls**		
	**Δ Mean ± SD**	** *t* **	** *p* **	**Δ Mean ± SD**	** *t* **	** *p* **
Weight (kg)	0.35 ± 3.04	0.636	0.530	1.40 ± 4.79	1.678	0.103
Height (cm)	2.81 ± 3.15	4.892	<0.001	3.48 ± 2.68	7.455	<0.001
Fat mass (%)	−3.53 ± 3.67	−5.901	<0.001	−3.40 ± 3.70	−5.288	<0.001
Fat mass (kg)	−1.66 ± 4.00	−2.270	0.031	−1.58 ± 3.70	−2.459	0.020
Fat-free mass (kg)	2.35 ± 2.72	4.731	<0.001	3.28 ± 3.16	5.958	<0.001
BMI (kg/m^2^)	−0.69 ± 1.84	−2.022	0.053	−0.56 ± 1.85	−1.660	0.108
WC (cm)	−3.53 ± 3.20	−4.542	<0.001	−3.64 ± 4.57	−3.730	0.001
WHtR	−0.12 ± 0.22	−2.058	0.060	−0.11 ± 0.18	−2.407	0.029
MBT forward (cm)	0.41 ± 0.77	2.934	0.006	0.51 ± 0.56	4.958	<0.001
MBT backward (cm)	0.79 ± 0.94	4.305	<0.001	0.64 ± 0.89	3.512	0.002
ST (kg)	3.08 ± 5.38	2.860	0.009	3.76 ± 4.12	4.182	<0.001
HG test for right hand (kg)	2.84 ± 5.45	2.020	0.063	0.59 ± 4.42	0.630	0.535
HG test for left hand (kg)	0.45 ± 3.78	0.432	0.674	1.24 ± 3.77	1.502	0.149
6 MWT (m)	69.03 ± 85.17	4.440	<0.001	51.28 ± 73.78	3.807	0.001
30 mt speed (s)	−0.64 ± 0.59	−5.823	<0.001	0.58 ± 0.48	6.450	<0.001
VB (cm)	2.59 ± 4.85	2.871	0.008	0.77 ± 4.06	1.035	0.309
HB (cm)	1.56 ± 4.40	1.774	0.089	1.96 ± 5.42	1.771	0.090
**(b)**						
	**Boys**			**Girls**		
	**Δ Mean ± SD**	** *t* **	** *p* **	**Δ Mean ± SD**	** *t* **	** *p* **
Weight (kg)	−1.67 ± 3.75	−1.337	0.218	−2.37 ± 3.39	−2.621	0.021
Height (cm)	1.5 ± 1.04	4.310	0.003	0.26 ± 0.95	1.046	0.314
Fat mass (%)	−4.89 ± 5.09	−2.881	0.020	−2.78 ± 2.53	−4.107	0.001
Fat mass (kg)	−3.12 ± 4.33	−2.161	0.063	−3.19 ± 2.36	−5.045	<0.001
Fat-free mass (kg)	3.57 ± 5.09	2.108	0.068	0.77 ± 3.21	0.898	0.386
BMI (kg/m^2^)	−1.12 ± 0.99	−3.388	0.010	−0.99 ± 1.26	−2.940	0.011
WC (cm)	−4.11 ± 2.62	−4.709	0.002	−4.00 ± 5.51	−2.719	0.018
WHtR	−0.29 ± 0.02	−5.942	<0.001	−0.03 ± 0.03	−2.828	0.014
Lat (kg)	9.28 ± 4.11	9.828	<0.001	7.65 ± 5.17	5.533	<0.001
Chest (kg)	10.65 ± 6.86	6.766	<0.001	8.08 ± 3.73	8.105	<0.001
Leg (kg)	38.79 ± 41.51	4.073	0.001	43.52 ± 31.76	4.940	<0.001
Legext (kg)	17.67 ± 12.57	6.126	<0.001	16.10 ± 7.73	7.509	<0.001
HG test for right hand (kg)	−7.58 ± 5.75	−3.485	0.013	−3.99 ± 9.65	−1.168	0.281
HG test for left hand (kg)	−6.86 ± 6.58	−2.759	0.033	−3.33 ± 7.86	−1.198	0.270
VB (cm)	1.78 ± 6.18	0.863	0.413	1.89 ± 3.66	1.550	0.160
HB (cm)	1.89 ± 3.06	1.852	0.101	0.78 ± 4.68	0.510	0.624
VO_2_ max	1.33 ± 4.28	0.934	0.378	1.02 ± 2.40	1.279	0.237

**Legend:** BMI = body mass index; WC = waist circumference; WHtR = waist-to-height-ratio; MBT = medicine ball trough test value; ST = Sargent test value; HG = Hand Grip test value; 6 MWT = six-minutes test value; 30 mt speed = 30 metres speed test value; VB = vertical bending test value; HB = horizontal bending test value; VO_2_ max = maximal oxygen consumption value. LAT = lat machine test value; CHEST = chest press test value; PRESS = leg press test value; LEGEXT = leg extension test value.

## Data Availability

The datasets used and/or analyzed during the current study are available from the corresponding author on reasonable request.

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
