# Peer review of "Effects of a Mixed Exercise Program on Overweight and Obese Children and Adolescents: A Pilot, Uncontrolled Study"

_ijerph, 2022, doi:10.3390/ijerph19159258_

Round 1
Reviewer 1 Report
The title and the aim of the work do not correspond.
I consider that your research design is not appropriate because you planned to compare two different populations (with different body composition changes due to puberty) who did not have the same exercise interventions: in time (children have 6 months and adolescents only 3 months) and modalities. Also, It is important that you consider the changes in puberty, in this sense I suggest not comparing children vs adolescents, just evaluate the efficacy of a mixed resistance-endurance exercise program per group.
If you have previous nutritional and psychological results, it would be interesting to discuss the effects of your exercise program on them or at least mention what was the nutritional intervention to which your groups were subjected.
How are you going to rule out whether the changes in body composition were due to the nutritional intervention, the exercise program, or puberty changes?
To define obesity in your groups, I suggest checking this work: Kuczmarski RJ, Ogden CL, Guo SS, et al. CDC growth charts for the United States: methods and development. Vital Health Stat 2000; 11: 1–190.
Author Response
We thank the reviewer for his/her comments.
Please see the file attached.

Reviewer 2 Report
Reviewer #1: IJERPH-1794193.
Evaluation of body composition and muscular strength in a group of overweight, obese young people: a pilot, uncontrolled study.
This paper assessed the effects of exercise and counseling interventions on two different groups (children and adolescent). Baseline and follow-up assessments were performed on each group, and statistical analysis was assessed to see if there were any statistical significance.
The overall manuscript is written well, and the reviewer can tell the authors spent a lot of time and effort on the writing, and design of the study. The main critiques of the manuscript relate to:
1) Methods
2.1 Participants: What specific BMI chart/scale did you use for the present study? Also, how would you classify someone as normal, overweight, and obese according to one younger than 18 years of age? Please elaborate in the manuscript, and provide a reference in the manuscript as well.
2.2.1 Exercise intervention for children:
Children were involved in a six-month exercise intervention program, with two sessions per week lasting 90 minutes each. Physical activities for this group were organized in small groups of min 6/max 12 participants. Two certified exercise specialists supervised all the training sessions, consisting of 15 minutes of warm-up, 50 minutes of calisthenics, mixed exercises to improve conditional and coordination skills, a group game phase (15 minutes), and a final stretching (10 minutes).
What were the specific calisthenics implemented over the program? Did they vary every week? Every training split? Further information is needed here. Same for mixed exercises and coordination skills. What types of final stretches were performed? How long were the stretches held for?
2.2.2. Exercise intervention for adolescents. For the adolescents’ group participated in a three-month exercise intervention, with two sessions per week, each lasting 90 minutes. Activities were organized for 5-6 participants/group, with one certified exercise specialist supervising. Each session included 60 minutes of cardiovascular workouts (using ergometers such as, i.e., Run 500, Recline 600 Xt Pro, and Top 600xt Pro and Synchro, Technogym, Cesena, Italy), 30 minutes of circuit training for muscular strength (using free lifts and isotonic machines, such as, i.e., Lat Machine, Chest Press, Leg Press, Technogym, Cesena, Italy).
Why did the adolescents only train for 3 months, while the children trained for 6 months? Why did the children use calisthenic training, while the adolescent used free weight and isotonic machines? Per the adolescent group, did any of them have previous resistance training experience, and that’s why there was a significant difference in strength between males and females in that age range? What were the specific sets, reps, rest periods that were used for the resistance training design for the adolescent group?
Author Response
We thank the reviewer for his/her comments.
Please see the attachment.

Reviewer 3 Report
Dear authors: With great respect, I make some considerations that I suggest should be addressed so that the paper can improve its quality for the reader.
*The abstract must be 200 words or less, contains 206
It is recommended that the keywords are not the same as the title
The methodology should be described in the abstract. Restructure since it does not contain it
line 32-37: Only describes physiological examples but mentions mental health, mentions some examples.
Line 42-49: Repetitive with the previous paragraph
Line 45-46: Your appointment and reference are needed
In relation to the introduction, it is necessary to mention whether or not it complies with the physical activity recommendations proposed by the WHO.
If the objective of the study is to verify the efficacy of a mixed program of strength-resistance exercises in anthropometric measurements and physical performance, evaluating the influence of sex and age, it is recommended to add in the introduction some other factor that may influence these. physical performance capabilities or on anthropometric measures such as maturation. The article Changes in strength and VO2max by physical training according to the state of maturation in children is included https://www.mdpi.com/2227-9067/9/7/938 where the topic was addressed, In addition to observing results according to the proposed training.
It is necessary to highlight the importance of the study, as well as the problems addressed to provide a solution through this document. Neither the hypothesis nor the relationship of the variables is mentioned.
It is recommended to restructure the introduction
materials and methods
Line 63: Explain why the sample was recruited from 2013 to 2017, how the program works. The sample is small since it was recruited over a long period of years, in addition the participants have common characteristics to be found in the population.
Line 70 the inclusion criteria are not mentioned, it is necessary to mention the adherence percentage in order to be taken into account in the program
Inscription
Explain why in the group of adolescents 23 participated in the anthropometric measurements and 33 in the Physical Performance Measures
The total of the mentioned sample of 138 participants does not match
Line
112 and 119: It is necessary to mention the intensity and progression of the exercise, adaptation is not mentioned, in addition to emphasizing whether the guidelines proposed by the WHO were followed, aspects for the work of strength, resistance, coordination and recreational aspects .
Line 130: The characteristics requested from the participants prior to the measurements are not mentioned.
Line 145: Was one or two evaluations carried out, was the average or maximum force taken into account? Was it produced as a continuous or categorical variable?
Line 151: The indicator is not VO2 max. Since it does not express the maximum value evaluated with the aforementioned test. It's just VO2
Line 152: Justify the use of the 6MWT, since other indirect methodologies such as PACER are mostly used for the child population. The validation of the instrument in children is not mentioned.
It is necessary to mention the connections between the 6MWT and a gold standard such as the gas analyzer.
About the methodology: It is necessary to mention with greater accuracy and precision how the measurements were carried out, the techniques, validations and justification of why these tests were used are not shown.
Results:
Table 1: It is not necessary to put the data of the m of the 6MWT since the indicator is the VO2 that is mentioned in the same table
In general, it would be important to mention the results by categories and not only do the analysis as a continuous or ratio variable, since if a higher value is presented but its category does not increase, it is not relevant at the health level, for example, advancing from a poor VO2 to an average VO2 for your age.
It is suggested to carry out the proposed analysis but through categories and show the size of the effect in each variable.
Discussion:
Line 240-242: The data they mention is controversial, it is recommended to expand the literature.
Line 248-250: Give a broader explanation of why these results were not obtained, since there are programs of shorter duration that obtain positive results in relation to the changes.
Line 274-289: Give an explanation of the results and not only the comparison with other authors
Line 308: Because a multidisciplinary work is mentioned if in the same limitations they mention that there was no nutritional or psychological advice
Conclusions: A conclusion cannot be generalized with such a small group.
I think the sample is too small to draw any firm conclusions. The n is not representative of the population.
Although the limitation of not having a control group is already mentioned, a justification is needed as to why it was not included in the study.

Author Response

(The authors gave the same response as above.)

Round 2
Reviewer 1 Report
1.- Define WHTR in the abstract
2. I suggest to focus the conclusion towards the main results related to the effects of the exercise program rather than the tools to evaluate the anthropometric characteristics and strength measurement.
3.- At what time of the day the participants' body weight measurement and air displacement plethysmography were carried out? In the morning or in the afternoon?
Reviewer 2 Report
Your revisions have increased the readability of the paper, and cleared up some confusion. My questions are in regards to the methods section, specifically the Exercise Intervention.
2.2.1. Exercise Intervention
With the creation of your Figure 2, clarification is needed with your sets and repetition scheme for your adolescent exercise creation. Additionally, what were the rest periods?
When you created the workouts, what reference were you using for that specific age group?
Please specify the exact order of the workout design.
Warm-up
Whats the exact exercise with sets, reps, intensity and rest period.
What's the cool-down and stretching protocol?
You mention HRR, were smart watches, or heart rate monitors worn during exercises, how were you able to track HR throughout those modes of training?
Reviewer 3 Report
Dear authors: Thank you for sharing your comments, however, there are still some points to strengthen, mainly in the conclusion. They are listed below:
• The abstract still exceeds the number of words
• The variables are not described in the methodology
• WC abbreviation not previously described
• In the results where it says "and some performance measures" it should be mentioned which ones since it shows the value of "p"
• Again abbreviations for BMI, WC and WHTR are not described
• Again, “some measures of muscular strength” are mentioned, which is not specific for a summary.
• The conclusion in the summary does not meet the general objective, it is not concluded in relation to gender or age.
• "with the combination of strength and dynamometric tests plus the study of body composition using aerial plethysmography methodology" is not stated in the objective
• Reference one does not correspond to what was stated in the sentence
• This explanation could well be exlciated in the methodology: Between 2010 and 2017 C.U.R.I.A.Mo. has carried out clinical activity mainly with adult patients (around 1000/year); In recent years, the treatment model has also been tested (and validated) in young subjects, who, however, were assigned a much smaller structure access number (around 50/year) than Adults.
• In point 2.2.1: Based on previous clinical experience, it is better not to mention clinical experience but to cite studies where exercise programs are described, stories such as: https://doi.org/10.3390/children9070938; . https://doi.org/10.17979/sportis.2021.7.3.8572; Analysis of body composition according to exercise modality in obese adults: Pilot study. Central European Journal of Sports Sciences and Medicine, 34, 87-95. These are some examples, but I suggest highlighting the exercises with previously published articles.
• It is necessary to strengthen the conclusion based on age and sex, since it is a fundamental part of the general objective.
